# Sustainability Assessment of the Utilization of CO₂ in a Dielectric Barrier Discharge Reactor Powered by Photovoltaic Energy

**Josep O. Pou** [1,*] **, Eduard Estopañán** [1] **, Javier Fernandez-Garcia** [1,2] **and Rafael Gonzalez-Olmos** [1]

1    IQS School of Engineering, Universitat Ramon Llull, Via Augusta 390, 08017 Barcelona, Spain
2    Department of Chemical Engineering, University College London (UCL), Torrington Place, London WC1E 7JE, UK
*    Correspondence: oriol.pou@iqs.url.edu; Tel.: +34-932672090

**Abstract:** The direct activation of diluted $CO_2$ in argon was studied in a co-axial dielectric barrier discharge (DBD) reactor powered by photovoltaic energy. The influence of the initial $CO_2$ and argon concentration on the $CO_2$ decomposition to form CO was investigated using a copper-based catalyst in the discharge zone. It was observed that the $CO_2$ conversion was higher at lower $CO_2$ concentrations. The presence of the diluent gas (argon) was also studied and it was observed how it has a high influence on the decomposition of $CO_2$, improving the conversion at high argon concentrations. At the highest observed energy efficiency (1.7%), the $CO_2$ conversion obtained was 40.2%. It was observed that a way to enhance the sustainability of the process was to use photovoltaic energy. Taking into account a life cycle assessment approach (LCA), it was estimated that within the best-case scenario, it would be feasible to counterbalance 97% of the $CO_2$ emissions related to the process.

**Keywords:** non-thermal plasma; $CO_2$ conversion; DBD; carbon capture and utilization (CCU)



## 1. Introduction

Nowadays, it is a priority to find and develop technologies to deal with $CO_2$ emissions in order to mitigate global warming [1]. New alternatives for carbon capture technologies have been studied and assessed [2–6], but the problem of which is the best alternative for $CO_2$ as a feedstock is still under study [7–10]. That is why $CO_2$ conversion to valuable products appears to be a great possibility; this leads to the reduction in fossil fuel dependence and global warming too.

The direct use of $CO_2$ has a great variety of applications, such as its relevance within soft drinks, to the scope of supercritical $CO_2$ as a solvent, and in other industrial areas, for instance, welding and refrigeration [7]. Direct dissociation of $CO_2$ and conversion into other value-added fuels and chemicals provides a potential route for efficient reduction in $CO_2$ emissions. Variable progress has been made to convert $CO_2$ into other value-added chemicals, such as $CO_2$ hydrogenation for the synthesis of methane, methanol, formaldehyde, etc. [8–10].

$CO_2$ conversion, decomposition, or activation into valuable products have been widely studied through thermo-chemical [11], photocatalysis [12–17], biological [18] or electrochemical [19,20] pathways, although these routes are not efficient from an energy perspective. Alternatively, non-thermal plasma (NTP) can operate under ambient conditions, so it can offer high energetic electrons which are able to initiate a highly endothermic chemical reaction under ambient temperature [21]. Different NTP approaches have been broadly studied such as corona discharge [22,23], gliding arc [24,25], microwave discharge [26,27], and dielectric barrier discharge (DBD) [28–30]. The latter has drawn more attention for the wide range of applications and its operability. Among several applications for the DBD approach, it has been successfully applied for volatile organic compounds treatment [31], wastewater treatment [32,33], reforming reaction [34], and methanol production [10,11,19,35].

NTP can operate at room temperature and atmospheric pressure while generating highly active electrons, with mean electron energy between 1 and 10 eV. This electron energy is the optimum range in order to activate molecular and atomic species and break chemical bonds [36]. NTP technologies are advantageous over thermal processes as reaction rates are higher and a steady state is achieved much faster [22]. On the other hand, the DBD reactor has the capacity to produce highly energetic electrons and uniform distribution discharges and is efficient in initiating chemical reactions under room conditions [37–39]. The combination of NTP with the main advantages of DBD reactors enables this system to be an ideal candidate for $CO_2$ utilization.

Therefore, NTP has high potential as an efficient $CO_2$ utilization process, as it can overcome the stability of $CO_2$ without the need for the high temperatures required in thermal catalytic processes. This facilitates quick start-up and shut-down, a promising feature that enables plasma technology powered by renewable energy to act as an efficient chemical energy storage [40].

In this work, the study of the conversion of $CO_2$ into CO through a DBD plasma reactor was developed by a sensitivity analysis for the $CO_2$ flowrate and the concentration of Ar that was used as a diluent; then, a sustainability assessment was considered for the suitability of using photovoltaic energy as an energy source to power the DBD plasma reactor.

## 2. Materials and Methods

### 2.1. Materials

The following reactants were used: Argon (Ar) (Carburos Metálicos; 99.9997%), Carbon dioxide ($CO_2$) (Carburos Metálicos; 99.999%) and the copper/zinc-based catalyst (Alfa Aesar; copper-based catalyst (pellets, 5.4 mm × 3.6 mm)) with a composition of $Al_2O_3$ 10%, CuO 64%, MgO 2% and ZnO 24%.

### 2.2. Experimental Setup

Figure 1 shows the experimental setup used in this work. Briefly, a cylindrical quartz tube, length: 250 mm; outer diameter: 25 mm; inner diameter: 24 mm, (Vidrasa, Ripollet, Barcelona, Spain) was used to generate the plasma discharge. A copper rod (Broncesval, Ripollet, Barelona, Spain) with an outer diameter of 5 mm was used as the internal electrode (anode) and the outer electrode (cathode) was a copper mesh of 160 mm, which was wrapped around the quartz tube, as shown in Figure 1. The resulting discharge gap was 10 mm. An amount of 40 g catalyst sandwiched between quartz wool was placed in the discharge area zone. NTP was generated by an AC high voltage power supply with a frequency, duty cycle, and voltage controller: Plasma Drive PVM500/DIDRIVE10 (Amazing1, Mont Vernon, NH, USA). The AC high-voltage power supply was connected to a photovoltaic (PV) system in order to provide the electricity needed to carry out the experimental tests. The PV system, which consisted of one solar panel Tecnosun 150 W, was placed outside to receive sunlight, using two batteries Rolls 6 V and 480 Ah and CC/AC inverter model Victron Energy Phoenix 12/1200. All the CC system worked at 12 V and was controlled by a regulator STECA PR 2020 regulator. In this work, the necessary energy to run the tests was provided directly from the PV panels. The energy consumption was continuously determined with a wattmeter. A multi-channel oscilloscope Promax 0D-610 100 MHz (Promax, L'Hospitalet de Llobregat, Barcelona, Spain) was used to monitor the voltage, frequency and current intensity during the experiences. The amount of $CO_2$ and Ar in the reactor was controlled with 2 flowmeters EL-FLOW® (Bronkhorst, Nijverheidsstraat, Ruurlo, The Netherlands).

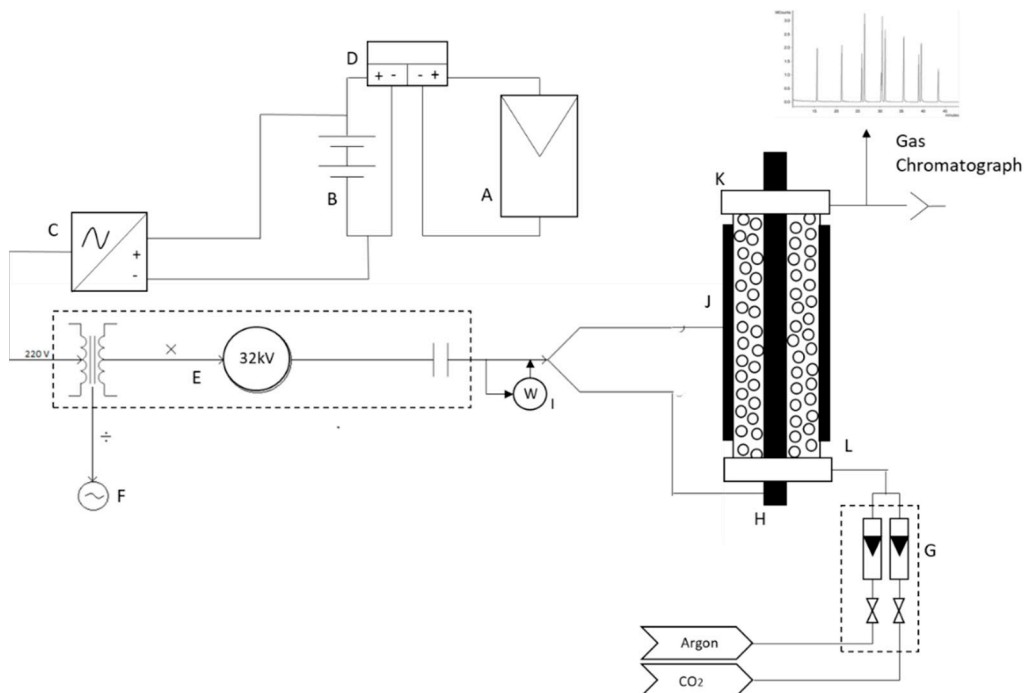

**Figure 1.** DBD system includes the photovoltaic power generation (solar panel (A), batteries (B), inverter (C), solar regulator (D)), electric signal amplifier (E), oscilloscope (F), volumetric flowmeters (G), copper rod inner electrode (H), wattmeter (I), copper mesh outer electrode (J) and PTFE reactor ends (K). The DBD reactor (L) is filled with catalyst pellets.

### 2.3. $CO_2$ Conversion into CO Experiments

All the experiments were performed at maximum voltage (32 kV), minimum frequency (20 kHz) and minimum duty changing flowrates of $CO_2$ and Ar. All the volumetric flowrates used in this study are given under normal conditions. The power used in all experiments was around 50 W. The total time for each experiment was 30 s once the steady state was reached. Samples were collected in Tedlar bags for further analysis. The samples were analyzed with a gas chromatograph (Agilent 7020A, Santa Clara, CA, USA) equipped with an Agilent HP-PLOT Molesieve 19095P-MS0 column and a thermal conductivity detector (TCD). The injection, oven, and detector temperatures were 100 °C, 60 °C, and 250 °C, respectively. All the experiments were carried out in triplicate and the average values are shown in the figures with the standard deviation as error bars.

### 2.4. Environmental Sustainability Assessment

The environmental sustainability of the process using PV energy was studied by calculating the carbon footprint. The carbon footprint was assessed with the software SimaPro 9.0 (Amersfoort, The Netherlands) using a Life Cycle Assessment (LCA) approach. For the LCA, the Ecoinvent v3.6 database and the ILCD 2011 Midpoint methodology were used considering two different scenarios. The first scenario considers the supply of electricity from the electrical network and the second scenario considers the electricity provided by a PV system.

To determine the percentage of the $CO_2$ emissions compensated ($EC_{co2}$), Equation (1) was used.

$$EC_{co2} \, (\%) = \frac{Con_{CO2} + P_{CO}}{E_{electricity}} \times 100 \tag{1}$$

where $Con_{CO2}$ (g $CO_2$-eq/h) was the $CO_2$ converted into CO in the DBD reactor and $P_{CO}$ (kg $CO_2$-eq/h) was the $CO_2$ emission that is related to the industrial production of the generated CO with a conventional process such hydrocarbon reforming. Finally, $E_{electricity}$ is the emission of $CO_2$ (g $CO_2$-eq/h) produced by the electrical consumption, which was

measured with a wattmeter. The emission factors to produce electricity (from the electrical network or PV systems) and for the industrial production of CO were extracted from Ecoinvent.

## 3. Results and Discussion

### 3.1. Influence of $CO_2$ Flowrate

Low $CO_2$ rates of decomposition have been found in the literature using pure $CO_2$, whereas the use of Ar has been reported to be a good option due to the use of a diluent gas (gas to be ionized) increases $CO_2$ conversion. When Ar is used, the main reaction pathway in the NTP zone involves Ar excitation and charge/energy transfer from excited Ar atoms to $CO_2$ molecules [41]. Mei et al. [28] reported that Ar dilution is beneficial over other gases such as $N_2$ and He. They observed that $CO_2$ conversion increases with increasing Ar concentrations. The reason is that by decreasing the dielectric strength of the gas mixture, more energy is available for $CO_2$ molecules in the discharge. In addition, the available electrons may excite $CO_2$ molecules contributing to reaching higher conversions due to the lower number of $CO_2$ molecules with respect to the Ar atoms. Mei et al. [28] also reported selectivities for $CO_2$ conversion into CO in high Ar concentrations between 90–98% that increased at increasing Ar concentration. For that reason, Ar was used as diluent gas in this work. At the studied conditions, when voltage was applied between the two electrodes in the DBD reactor, NTP discharges were produced within the plasma reactor, as Figure 2 shows. It was observed randomized discharge points within the DBD plasma reactor that continuously changed. The NTP was fast generated or stopped by switching on or off the AC high voltage power supply, quickly reaching the steady state as suggested by Pou et al. [40].

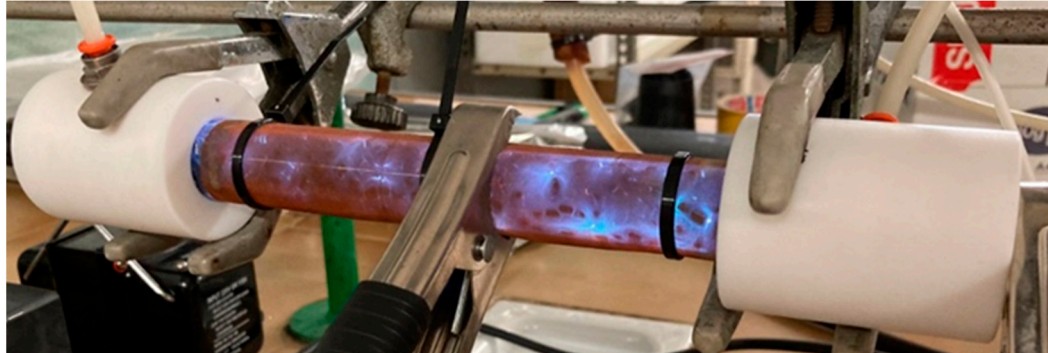

**Figure 2.** Picture of the NTP discharges across the DBD plasma reactor.

Two important outcomes to check the performance of the reaction are the $CO_2$ conversion and the energy efficiency ($\eta$). The $CO_2$ conversion (%) was determined with the following equation:

$$CO_2 \text{ Conversion } (\%) = \frac{C_{CO2,in} - C_{CO2,out}}{C_{CO2,in}} \times 100 \qquad (2)$$

where $C_{CO2,out}$ is the molar concentration (%) of $CO_2$ in the outlet stream and $C_{CO2,in}$ is the molar concentration (%) of $CO_2$ in the inlet stream of the DBD reactor. From the chromatography analysis, it was found that a selectivity of almost 100% for CO was reached for all the tests (no other peaks different from CO or $CO_2$ were observed). From these results, it was considered that CO was the major product and the stoichiometric conversion of $CO_2$ into CO was achieved. So, the conversion and yield obtained were almost the same. This point has also been reported in previous works [28,40] where selectivities to CO higher than 95% were observed. Carbon deposition over the catalyst was not detected after $CO_2$ decomposition with NTP. It must be highlighted that the CO selectivity did not present any dependence on other parameters studied in this work, such as $CO_2$ or Ar flowrates. Other works shown in Table 1 reported selectivities ranging from 48 to 96%. Therefore, the influence of plasma operational parameters on the NTP process will only be discussed in

terms of $CO_2$ conversion and energy efficiency. The energy efficiency of the NTP process was calculated using the following equation:

$$\eta(\%) = \frac{F_{CO2} \cdot CO_2 Conversion \cdot \Delta H_{CO}}{60 \times 22.4 \cdot P_E} \times 100 \tag{3}$$

where $F_{CO2}$ is the volumetric flowrate of $CO_2$ in the inlet stream (mL/min), $\Delta H_{CO}$ is the reaction enthalpy of pure $CO_2$ decomposition into CO (283.1 kJ/mol [42]) and $P_E$ is the electric power used to carry out the reaction (W).

**Table 1.** Comparison of DBD plasma-assisted conversion of $CO_2$.

| Flowrate (mL/min) | Diluent Gas | Power (W) | Packed Material | $CO_2$ Conv. (%) | CO Select. (%) | CO Yield (%) | η (%) | Ref |
|---|---|---|---|---|---|---|---|---|
| 30 | Ar | 2.4 | Glass beads | 19.5 | 86 | 16.8 | 17.0 | [39] |
| 150 | - | 55 | Molecular sieves 5A | 25 | 63 | 15.8 | - | [44] |
| 60 | - | 28 | BaTiO$_3$ | 38.3 | - | - | 17.0 | [45] |
| 50 | - | 50 | BaTiO$_3$ | 28 | 96 | 26.9 | 7.2 | [46] |
| 30 | - | 2.4 | 5% ZnO + g-C3N4 | 12 | 70 | 8.4 | 31.1 | [16] |
| 30 | - | 2.2 | 15% CuO/Al$_2$O$_3$ | 15.7 | 48 | 7.5 | 45.2 | [43] |
| 30 | - | 2.2 | 15% CuO/CeAl | 13.5 | 59 | 7.8 | 38.9 | [43] |
| 9 | Ar | 50 | CuO/ZnO/Al$_2$O$_3$ | 40.2 | >99 | 40.2 | 1.7 | This work |

The effect of the $CO_2$ flowrate on the $CO_2$ conversion and energy efficiency was evaluated. In Figure 3, it can be observed how the $CO_2$ conversion decreased by increasing the $CO_2$ flowrate in the inlet. According to previous research [40], it has been proved that the generated NTP is more uniform at high Ar concentrations. In addition, at lower $CO_2$ flowrates, the residence time in the reactor was higher and this would increase the probability that a NTP discharge could affect a $CO_2$ molecule. This behavior has also been described using a fluidized bed NTP reactor with $Cu/\gamma$-$Al_2O_3$ powder-based catalyst [40]. The maximum $CO_2$ conversion in this work was 74.2% and it was obtained at the lowest $CO_2$ flowrate. This conversion was much higher than the one obtained by Ray et al. (maximum conversion of 15.7%) utilizing a packed DBD plasma reactor with Ni and Cu oxide supported $\gamma$-$Al_2O_3$ as catalyst [43]. Other works (Table 1) reported conversions ranging between 12–38.3%. In addition, the maximum conversion obtained in this work is much higher than the previous results obtained by our research group with a fluidized bed reactor and a $Cu/\gamma$-$Al_2O_3$ catalyst (a maximum conversion of 40% was reached) [40].

The $CO_2$ flowrate shows an effect on the energy efficiency of the process, as can be observed in Figure 4. The maximum $CO_2$ energy efficiency obtained in this study was 1.55%, using an Ar flowrate of 1 L/min and a $CO_2$ flowrate of 9 mL/min. This value was similar to the results obtained with a fluidized bed reactor (1.3–2.0%) [40]; however, still less efficient than other works published in the literature working with DBD reactors that obtained energy efficiencies ranging from 7 to 45.2% (Table 1). The main reason to explain these differences in energy efficiency is caused by the different operational conditions (use of diluent gas, $CO_2$ flowrate, catalyst, and power discharge). Probably, the amount of energy applied (power discharge) related to the $CO_2$ flowrate used is too high compared with the literature. When the $CO_2$ flowrate increased too much, the energy efficiency was affected by a lower $CO_2$ conversion, as observed previously in Figure 3.

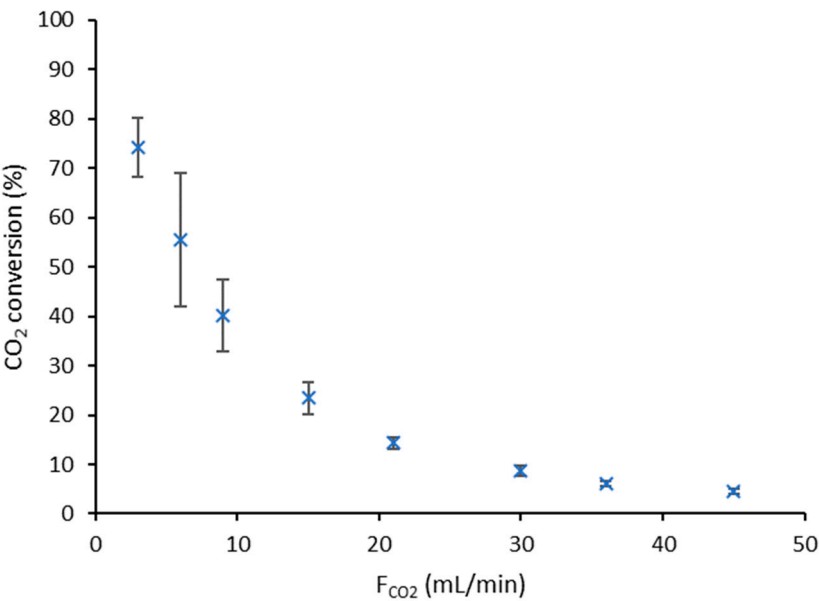

**Figure 3.** Effect of $CO_2$ flowrate on the $CO_2$ conversion. Conditions: 1 L/min of Ar, 50 W, 20 kHz and minimum duty cycle.

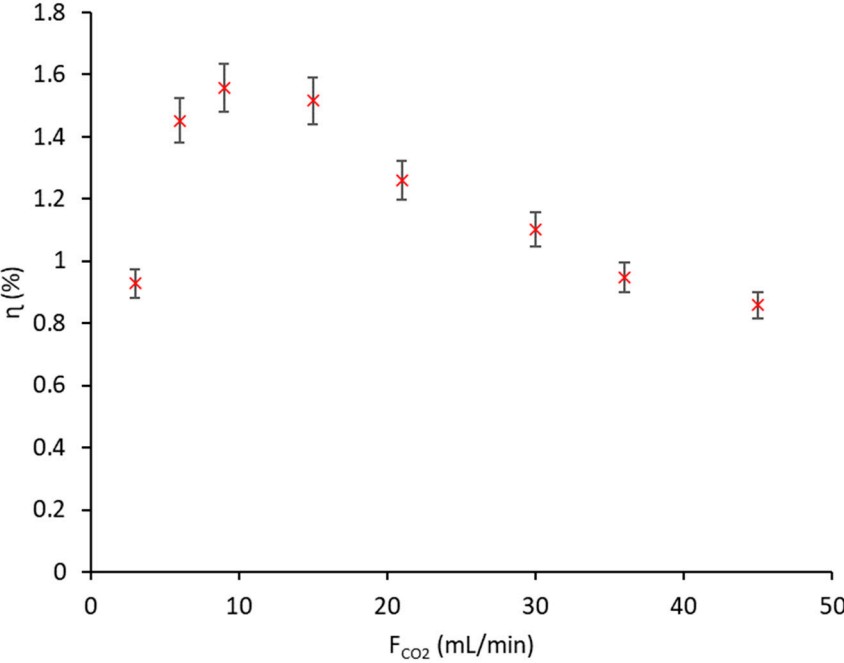

**Figure 4.** Effect of $CO_2$ flowrate on the energy efficiency. Conditions: 1 L/min of Ar, 50 W, 20 kHz, and minimum duty cycle.

The formation of active species in the DBD reactor is entirely dependent on the initial concentration of reactants and the specific input energy (SIE). The SIE (kJ/L) was calculated with the following equation:

$$SIE = \frac{P_E \cdot 60}{F_{CO2}} \qquad (4)$$

With a catalyst, not only the concentration of the reactants but also the reduced electric field enhances inside the plasma zone, which in turn generates more chemically reactive

species and significantly contributes to the activation of $CO_2$. The conversion rate of $CO_2$ was determined from Equation (5) [16].

$$\ln\left(\frac{C_{CO2,\ in}}{C_{CO2,\ out}}\right) = SIE \cdot k + c \tag{5}$$

The conversion rate is expressed by k, and c stands for the intercept. Figure 5 reports the lineal increment of $\ln(C_{CO2},in/C_{CO2},out)$ as a function of SIE. The decomposition rate observed was of 0.0014 L/kJ. This value was lower than the value obtained by Ray et al. [43], who obtained a maximum conversion rate of 0.02889 L/kJ using 15% $CuO/Al_2O_3$ catalyst and without Ar dilution. They worked with lower SIE values which can also affect the conversion rate. So, it is plausible that the difference between conversion rates is caused by differences in the operational conditions.

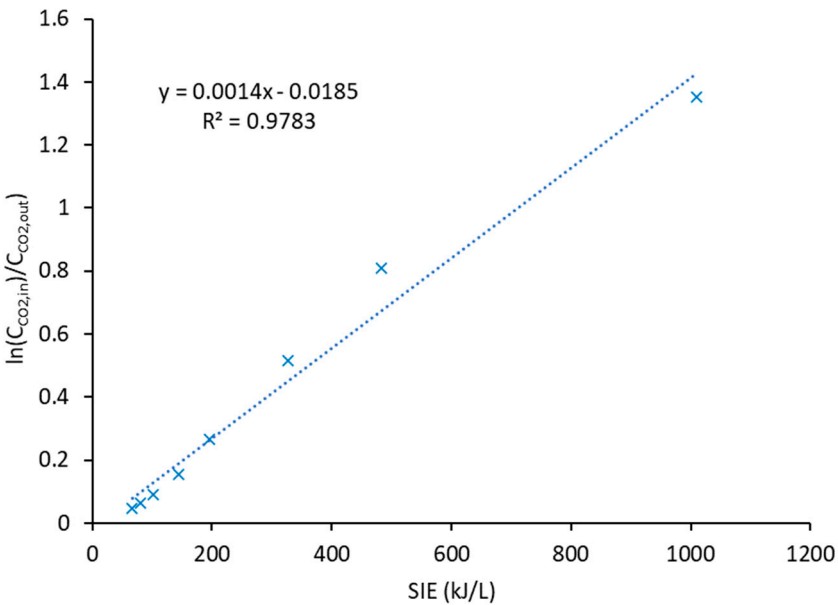

**Figure 5.** Rate of reaction as a function of SIE (linear fit).

### 3.2. Influence of Argon Flowrate

In the previous section, it was revealed that at lower $CO_2$ flowrates the residence time in the DBD plasma reactor was higher, thus, improving $CO_2$ conversion. Another important variable of the process is the use of Ar as a diluent gas. For that reason, it was studied the influence of the Ar flowrate on the $CO_2$ conversion into CO. This study was also carried out at different $CO_2$ flowrates (so at different $CO_2$ concentrations), and the results are shown in Figure 6.

Figure 6 shows how the conversion increases linearly with increasing Ar flowrates as reported in other works [39,40]. During the experiments, it was observed that at higher Ar flowrates, the NTP generated was more uniform and with a higher number of discharges inside of the reactor. This explains why at a high Ar flowrate the conversions obtained are higher.

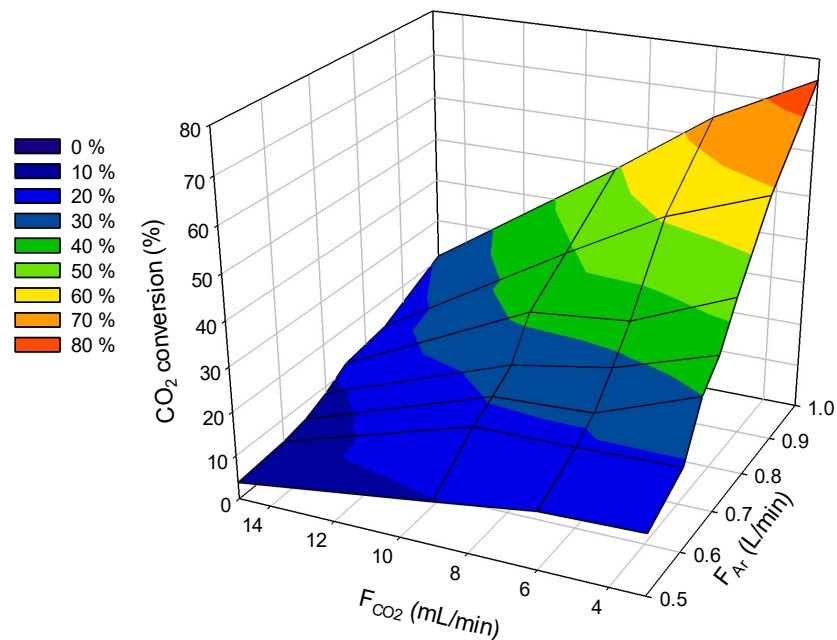

**Figure 6.** Effect of Ar flowrate on the $CO_2$ conversion. Conditions: 50 W, 20 kHz and minimum duty cycle. The experiments were performed at different $CO_2$ flowrates.

### 3.3. Environmental Impact Using Photovoltaic Energy to Power the DBD Reactor

The assessment of the $EC_{CO_2}$ was performed considering two different scenarios. The first scenario uses the electricity mix of Spain to run the DBD reactor, whereas the second scenario considers the current configuration of this work, supplying energy with a PV system. In Figure 7, it can be seen how the first scenario represents a lower compensation of $CO_2$ emissions (between 7–13%). It can be noticed that it shows a similar tendency to the energy efficiency plot (Figure 4), as they are correlated; therefore, with this first scenario, the emissions derived from the electricity mix are much higher than those which can be counterbalanced by the $CO_2$ captured and the CO produced during the DBD reaction. According to the Ecoinvent database, the emission factors related to the consumption of 1 kWh with the electricity mix of Spain and the production of 1 kg of industrial CO would be estimated as 0.131 kg of $CO_2$-eq and 1.51 kg of $CO_2$-eq, respectively.

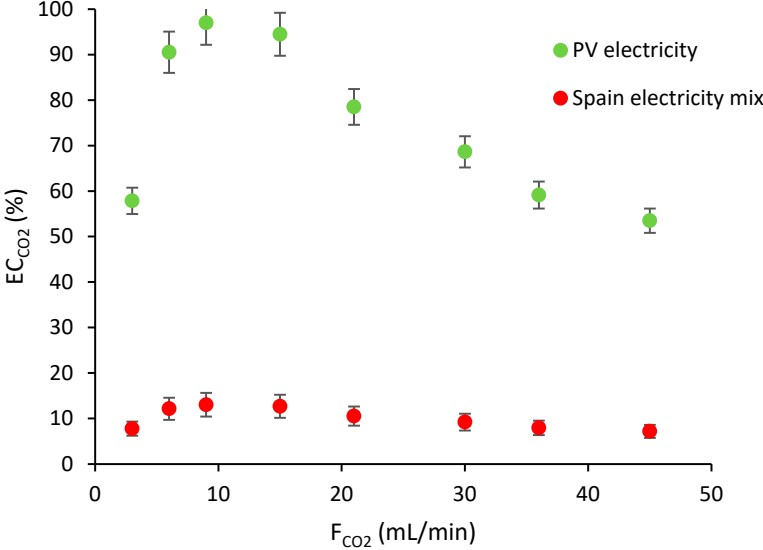

**Figure 7.** Effect of $CO_2$ flowrate and the type of electricity source on the carbon emissions compensated. Conditions: 1 L/min of Ar, 50 W, 20 kHz, and minimum duty cycle.

The optimal experiment (highest energy efficiency) was at a $CO_2$ flowrate of 9 mL/min. In this experiment, the $CO_2$ converted to CO ($C_{CO2}$) was 0.43 g $CO_2$-eq/h, whereas the emission avoided by the production of CO ($P_{CO}$) was 0.41 g $CO_2$-eq/h. The energy consumption was 0.11 kWh per each gram of $CO_2$ transformed into CO. The emission related to the electricity consumption ($E_{electricity}$) considering the electricity mix of Spain was 6.42 g $CO_2$-eq/h.

Taking into account the PV system scenario, the $EC_{CO2}$ rose up to 97% (Figure 7). This shows how the combination of the NTP process with PV systems significantly enhances the sustainability perspectives of the process. This increase would be justified by the lower emission factors related to the production of 1 kWh using PV panels (Ecoinvent database $\rightarrow$ 0.018 kg of $CO_2$-eq). There is a difference of one order of magnitude lower than the emission factor of the electricity from the electrical grid. The emission related to the electricity consumption ($E_{electricity}$) in the optimal experiment (9 mL/min of $CO_2$) using the PV system was 0.86 g $CO_2$-eq/h. That is why the obtained results clearly evidenced that the combination of the DBD reactor with renewable energy significantly enhances the life cycle assessment of the process, reaching almost a complete compensation of the $CO_2$ emissions with a more sustainable perspective.

## 4. Conclusions

A cylindrical packed-bed DBD plasma reactor was used for the conversion of $CO_2$ into CO using a copper and zinc-based catalyst, copper electrodes, Ar as a diluent gas, and photovoltaic energy as an electricity source. Parameters such as $CO_2$ flowrate in the inlet gas stream and Ar flowrate were assessed. The main findings of this study are stated below:

- The concentration of $CO_2$ in the inlet of the reactor is an important variable. The $CO_2$ conversion is higher at lower $CO_2$ concentrations. An increase in $CO_2$ concentration causes a major decline in $CO_2$ conversion.
- The maximum $CO_2$ conversion was 74.2%, using an Ar flowrate of 1 L/min and a $CO_2$ flowrate of 3 mL/min, applying 50 W, a frequency of 20 kHz, and a minimum duty cycle.
- The presence of the diluent gas (Ar) has a strong influence on the decomposition of $CO_2$. It was observed that at higher Ar concentrations, the conversion improved.
- The use of photovoltaic energy increases the sustainability of the process. Using an LCA approach, it was estimated, for the decomposition reaction, that, with the best conditions obtained in this study, it would be possible to compensate 97% of the $CO_2$ emissions related to the process.

**Author Contributions:** Conceptualization, J.O.P. and R.G.-O.; methodology, J.O.P.; investigation, E.E., J.O.P. and R.G.-O.; resources, J.O.P.; writing—original draft preparation, J.O.P., J.F.-G. and R.G.-O.; writing—review and editing, J.O.P., J.F.-G. and R.G.-O.; supervision, R.G.-O.; project administration, J.O.P. and J.F.-G.; funding acquisition, J.O.P. and J.F.-G. All authors have read and agreed to the published version of the manuscript.

**Funding:** This research was funded by La Caixa grant number (2018-LC-13) and AGAUR (2021 BP 0029).

**Institutional Review Board Statement:** Not applicable.

**Informed Consent Statement:** Not applicable.

**Data Availability Statement:** Not applicable.

**Acknowledgments:** GESPA group has been recognized as a Consolidated Research Group by the Catalan Government with code 2017-SGR-1016. The authors acknowledge La Caixa and AGAUR (Catalan Government) for the funding.

**Conflicts of Interest:** The authors declare that they have no known competing financial interests or personal relationships that could have appeared to influence the work reported in this paper.

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
