# Peer review of "Sustainability Assessment of the Utilization of CO2 in a Dielectric Barrier Discharge Reactor Powered by Photovoltaic Energy"

_processes, doi:10.3390/pr10091851_

Round 1
Reviewer 1 Report
The authors presented a study on PV-powered dielectric barrier discharge reactor to reduce CO2 to CO. The idea seems plausible, but the efficiency is relatively low compared to other literature reports. These results need to be justified, and the manuscript has to be improved before it can be accepted.
In Figure 3, it appears at a low flow rate of less than 10 ml/min there are more uncertain as indicated by error bars. What makes the data more scattered?
- The reported efficiency of 1.7% is one magnitude lower than the typical values from the literature. What causes the drastic difference?
- Why is the conversion rate so low compared to Ray et al. [47]?
- Several sentences that ran for 4-5 lines can be rephrased to improve the readability.
Author Response
Reviewer 1:
General comment: The authors presented a study on PV-powered dielectric barrier discharge reactor to reduce CO2 to CO. The idea seems plausible, but the efficiency is relatively low compared to other literature reports. These results need to be justified, and the manuscript has to be improved before it can be accepted
AUTHORS REPLY: We agree with the reviewer that the efficiency results need to be justified and we have changed this part of the paper as suggested here and in the points below.
1) In Figure 3, it appears at a low flow rate of less than 10 ml/min there are more uncertain as indicated by error bars. What makes the data more scattered?
AUTHORS REPLY: We fully understand the reviewer’s concern regarding the uncertainty indicated with the error bars. As explained in the methodology section, our experiments were performed by triplicate. The reason to observe higher error bars at CO2 flow rates lower than 10 mL/min is due to these experiments were carried out in the low range of the manometer used, causing a slightly error as observed in Figure 3. For that reason and to have a good experimental trend of the influence of the CO2 flowrate all the experiments were carried out by triplicate. On the other hand, the errors at these low CO2 flowrates are lower than 8 % except in the case of the experimental point around 5 mL/min where the error bars are around 12%. So, we could say that these errors are acceptable.
2) The reported efficiency of 1.7% is one magnitude lower than the typical values from the literature. What causes the drastic difference?
AUTHORS REPLY: The main reason to explain the differences between the efficiency obtained in our system compared to the literature are the experimental conditions used. The main focus of the work was to study from a life cycle assessment approach if the use of plasma to reduce CO2 could be environmentally sustainable if renewable energy is used. The optimization of the system was out of the scope, and this is something that we will explore in the future. The differences in the energy efficiencies between our work and other published in the literature are caused by two main reasons: The first one is the use of argon as diluent gas (only one work of Table 1 used argon as diluent gas) and also the high power discharge used in our experiments probably due to our high voltage power supply is oversized for the scale of the experimental reactor. According to the reviewer concern we have included the following sentence in the new version of the manuscript.
Page 6, line 198-202: The main reasons to explain these differences in the energy efficiency is caused by the different operational conditions (use of diluent gas, CO2 flow rate, catalyst and power discharge). Probably, the amount of energy (power discharge) applied related to the CO2 flowrate used is too high comparing with the literature.
3) Why is the conversion rate so low compared to Ray et al. [47]??
AUTHORS REPLY: Again, as in the last comment, the differences with the conversion rate of our work and Ray et al. [47] is explained by the fact of the experimental conditions. Ray et al they don’t use argon as diluent gas what can affect the efficiency in the use of energy to convert CO2 into CO. The second fact is related to the SIE used in the different experimental set-ups. In our experiments SIEs were one order of magnitude higher what probably caused a worse utilization of the applied energy. Ray et al worked with a CO2 flow rate of 30 mL/min (without argon dilution) and used a power discharge of 2.2 W while we worked at a much higher power discharge (50 W) resulting in higher SIE values. We have included a sentence in the manuscript in order to explain the plausible reasons that explain this difference.
Page 7, line 226-228: They worked with lower SIE values which can also affect to the conversion rate. So, it is plausible that the difference between conversion rates is caused by differences in the operational conditions.
4) Several sentences that ran for 4-5 lines can be rephrased to improve the readability.
AUTHORS REPLY: We agree with the reviewer and several long sentences with 4-5 lines have been rephrased to improve the readability. The sentences rephrased are shown below.
Page 2, line 48-56: Thus, the combination of NTP technology, which is advantageous over thermal processes as reaction rates are higher and steady state is achieved much faster [22], and the main advantages of DBD reactor with the capacity to produce high energetic electrons, uniform distribution discharges and efficient initiator of chemical reactions under room conditions [37–39], enables this system to be optimized for the best performance in CO2 utilization.
Rephrased: NTP technology is advantageous over thermal processes as reaction rates are higher and steady state is achieved much faster [22]. On the other hand, the DBD reactor has the capacity of producing high energetic electrons, uniform distribution discharges and efficient initiating chemical reactions under room conditions [37-39]. The combination of NTP with the main advantages of DBD reactors enables this system to be an ideal candidate for CO2 utilization.
Page 4, line 131-135: Considering pure CO2, low rate of decomposition has been found on literature whereas the use of Ar has been evidenced to be a good option as diluent gas to support the increase of the CO2 conversion, as the main reaction pathway in the NTP zone involves argon excitation and charge/energy transfer from excited argon atoms to CO2 molecules [41].
Rephrased: Low CO2 rate of decomposition has been found in the literature using pure CO2 whereas the use of Ar has been evidenced to be a good option as diluent gas to increase of the CO2 conversion. This is caused because the main reaction pathway in the NTP zone involves argon excitation and charge/energy transfer from excited argon atoms to CO2 molecules [41].
Page 4, line 156-160: From the chromatography analysis it was found the total and complete selectivity of almost 100% for CO was reached for every test (no other peaks different to CO were observed) which indicated that CO was the major product from CO2 decomposition and the stoichiometric conversion of CO2 into CO was achieved.
Rephrased: From the chromatography analysis it was found that a selectivity of almost 100% for CO was reached for all the test (no other peaks different to CO were observed). From these results, it was considered that CO was the major product and the stoichiometric conversion of CO2 into CO was achieved.

Reviewer 2 Report
In this work, a cylindrical packed-bed DBD plasma reactor was used for the conversion of CO2 into CO using a copper and zinc-based catalyst, copper electrodes, Ar as diluent gas and photovoltaic energy as electricity source. This work can be published after addressing the following issues.
(1) Any products besides the CO after CO2 decomposition.
(2) For the effect of CO2 flowrate on the energy efficiency. Why it has an optimum efficiency at around 10 ml/min
(3) In table 1, the comparison of DBD plasma assisted CO2 conversion. The higher efficiency in this work is probably attributed to the relatively low flow rate compared to previous work. The comparison should be made under the similar boundary conditions.
(4) The presence of the diluent gas (argon) has a high influence over the decomposition of CO2. Underlying reasons are needed.
Author Response
Reviewer 2:
General comment: In this work, a cylindrical packed-bed DBD plasma reactor was used for the conversion of CO2 into CO using a copper and zinc-based catalyst, copper electrodes, Ar as diluent gas and photovoltaic energy as electricity source. This work can be published after addressing the following issues.
AUTHORS REPLY: We thank the reviewer for his/her positive evaluation on our work.
1) Any products besides the CO after CO2 decomposition.
AUTHORS REPLY: Thank you the reviewer for this comment. According to our chromatography analysis, we did not find any other different peak to CO and CO2 that could indicate the presence of other compounds. Carbon deposition over the catalyst was also not observed after the reaction. In order to clarify that, we have modified a sentence (current lines 155-163) that was included in the original submission. “From the chromatography analysis it was found that a selectivity of almost 100% for CO was reached for all the test (no other peaks different to CO or CO2 were observed). From these results, it was considered that CO was the major product and the stoichiometric conversion of CO2 into CO was achieved. So, the conversion and yield obtained were almost the same. This point has also been reported in previous works [43,44] where selectivities to CO higher than 95% were observed. Carbon deposition over the catalyst was not detected after CO2 decomposition with NTP.”
2) For the effect of CO2 flowrate on the energy efficiency. Why it has an optimum efficiency at around 10 ml/min
AUTHORS REPLY: The reason to observe an optimum is that if the CO2 flow rate is increased too much the energy efficiency is affected by a lower CO2 conversion as observed in Figure 3. This was explained in the original submission (current Lines 202-203).
3) In table 1, the comparison of DBD plasma assisted CO2 conversion. The higher efficiency in this work is probably attributed to the relatively low flow rate compared to previous work. The comparison should be made under the similar boundary conditions.
AUTHORS REPLY: The energy efficiencies (η) are lower in our work. We suppose that the reviewer in this comment refers to the conversion (C) of Table 1 that in our work (40.2%) has a higher value than in the other works (ranging from 12%-38%). We agree with the reviewer that probably the observed higher efficiencies are caused by the lower CO2 flow rates used in our study. The CO2 flow rates reported in the literature vary in a wide range (from 30 to 150 mL/min) but also the catalysts (glass beads, zeolites, Ti-based catalyst, Zn-based catalyst, and Cu-based catalyst) and the power discharge (ranging from 2.4 to 55 W) used what makes difficult to do a fair comparison. What we try to do with this section and Table 1 is to show the reported optimum results obtained in the different works with DBD in terms of conversion and energy efficiency independently of the conditions which are in general quite different. Also, it is important to remark that the main objective of this work is to assess the environmental sustainability of a DBD reactor powered by PV panels and the optimization of the system was out of the scope of this work.
4) The presence of the diluent gas (argon) has a high influence over the decomposition of CO2. Underlying reasons are needed.
AUTHORS REPLY: We thank the reviewer for his/her appreciation. In the case of pure CO2 low rate of decomposition has been reported while it has been proved that the use of argon as diluent gas help to increase the CO2 conversion because the major reaction pathway in the non-thermal plasma zone involves argon excitation and charge/energy transfer from excited argon atoms to CO2 molecules [1-2]. During the experiments and as it was observed in previous research [1], it was observed that with high argon concentration the non-thermal plasma generated was more uniform and with a higher number of discharges inside of the reactor. So, at higher argon flowrates a higher number of non-thermal plasma discharges were observed contributing to a higher CO2 conversion as shown in Figure 6. An additional sentence in the manuscript has been included clarifying this aspect.
[1] Pou et al. CO2 reduction using non-thermal plasma generated with photovoltaic energy in a fluidized reactor. Journal of COâ‚‚ Utilization 27 (2018) 528–535
[2] H. Matsumoto, S. et al. Profiles of carbon dioxide decomposition in a dielectric-barrier discharge-plasma system, Bull. Chem. Soc. Jpn. 72 (1999) 2567–2571.
Page 7-8, line 237-242: Figure 6 shows how the conversion increases linearly with increasing Ar flowrates as reported in other works [40,42]. During the experiments, it was observed that ats higher Ar flowrate, the NTP generated was more uniform and with a higher number of discharges inside of the reactor. This explains why at high Ar flowrates the conversion is higher.

Reviewer 3 Report
The manuscript studied the “Sustainability assessment of the utilization of CO2 in a dielectric barrier discharge reactor powered by photovoltaic energy”, the influence of the initial CO2 and argon concentration on the CO2 decomposition to form CO was investigated using a copper-based catalyst in the discharge zone, it was observed how the CO2 conversion was higher at lower CO2 concentrations. The whole review is detailed and comprehensive. Therefore, I think that this work is appropriate for publication after minor revision as follows:
1. There are many minor errors in the text that I hope the author can check and correct carefully. For example, Figure 3 on line 204 should actually correspond to Figure 4; A similar mistake is that the CO2 flow rate is sometimes ml/s, sometimes ml/min. The author should check for similar unit labeling errors and standardize the whole text.
2. In line 190 of Section 3.1, the author compares other works to show that the conversion rate in this work is relatively high, but the factors such as gas flow rate, dilution gas and power are different. Can this form an effective comparison? Whether it is necessary to compare the conversion rate under the condition of control variables.
3. The grammar check and usage of this manuscript need to be modified, it is also suggested that the authors check the similar grammar problems in the whole paper.
Author Response
Reviewer 3:
General comment: The manuscript studied the “Sustainability assessment of the utilization of CO2 in a dielectric barrier discharge reactor powered by photovoltaic energy”, the influence of the initial CO2 and argon concentration on the CO2 decomposition to form CO was investigated using a copper-based catalyst in the discharge zone, it was observed how the CO2 conversion was higher at lower CO2 concentrations. The whole review is detailed and comprehensive. Therefore, I think that this work is appropriate for publication after minor revision as follows:
AUTHORS REPLY: We thank the reviewer for his/her positive evaluation on our work.
1) There are many minor errors in the text that I hope the author can check and correct carefully. For example, Figure 3 on line 204 should actually correspond to Figure 4; A similar mistake is that the CO2 flow rate is sometimes ml/s, sometimes ml/min. The author should check for similar unit labeling errors and standardize the whole text.
AUTHORS REPLY: We thank the reviewer for observing these typographic errors. We have corrected all of them and also, we have done an extensive check for similar unit labeling errors and we have corrected them. Regarding the citation of Figure 3 in line 204 is right because in this part we are explaining that the trend observed in Figure 4 is related to the decrease of the conversion shown in Figure 3. We have corrected the units of the flowrates in the whole manuscript. For CO2 are given in mL/min and for Ar in L/min.
2) In line 190 of Section 3.1, the author compares other works to show that the conversion rate in this work is relatively high, but the factors such as gas flow rate, dilution gas and power are different. Can this form an effective comparison? Whether it is necessary to compare the conversion rate under the condition of control variables.
AUTHORS REPLY: We agree with the reviewer that the comparison should be done with very similar conditions but regrettably they are very different in the works reported in the literature. The CO2 flowrates reported in Table 1 vary in a wide range (from 30 to 150 mL/min) but also the catalysts (glass beads, zeolites, Ti-based catalyst, Zn-based catalyst, and Cu-based catalyst) and the power discharge (ranging from 2.4 to 55 W) used what makes difficult to do a fair comparison. What we try to do with this section and Table 1 is to show the reported optimum results obtained in the different works with DBD in terms of conversion and energy efficiency independently of the conditions which are in general quite different.
3) The grammar check and usage of this manuscript need to be modified, it is also suggested that the authors check the similar grammar problems in the whole paper.
AUTHORS REPLY: Thank you to the reviewer for his/her appreciation. We have checked thoughtfully the grammar of the whole paper.
